# Is Slovakia Almost a Hepatitis D Free Country?

**DOI:** 10.3390/v15081695

**Published:** 2023-08-05

**Authors:** Pavol Kristian, Ivana Hockicková, Elena Hatalová, Daniela Žilinčanová, Marek Rác, Veronika Bednárová, Patrícia Denisa Lenártová, Sylvia Dražilová, Ľubomír Skladaný, Ivan Schréter, Peter Jarčuška, Monika Halánová

**Affiliations:** 1Department of Infectology and Travel Medicine, Faculty of Medicine, Louis Pasteur University Hospital, Pavol Jozef Safarik University, 041 90 Kosice, Slovakia; pavol.kristian@upjs.sk (P.K.); patricia.denisa.lenartova@upjs.sk (P.D.L.); ivan.schreter@upjs.sk (I.S.); 2Department of Epidemiology, Faculty of Medicine, Pavol Jozes Safarik University, 040 11 Kosice, Slovakia; elena.hatalova@upjs.sk (E.H.); veronika.bednarova@upjs.sk (V.B.); monika.halanova@upjs.sk (M.H.); 3II. Department of Internal Medicine, HEGITO Division Hepatology, Gastroenterology and Liver Transplantation, F. D. Roosevelt University Hospital, Slovak Medical University, 974 01 Banska Bystrica, Slovakia; daniela.zilincanova@gmail.com (D.Ž.); lubomir.skladany@gmail.com (Ľ.S.); 4Department of Internal Medicine, Faculty Hospital, 949 01 Nitra, Slovakia; marekrac@fnnitra.sk; 52nd Department of Internal Medicine, Faculty of Medicine, Louis Pasteur University Hospital, Pavol Jozef Safarik University, 041 90 Kosice, Slovakia; sylvia.drazilova@upjs.sk (S.D.); peter.jarcuska@upjs.sk (P.J.)

**Keywords:** hepatitis D, prevalence, anti-HDV antibody, HDV RNA

## Abstract

Background: It is assumed that the prevalence of hepatitis D in HBsAg-positive individuals reaches 4.5–13% in the world and on average about 3% in Europe. Data from several European countries, including Slovakia, are missing or are from an older period. Methods: We analyzed all available data on hepatitis D from Slovakia, including reports from the Slovak Public Health Authority and the results of one prospective study, and three smaller surveys. The determination of anti-HDV IgG and IgM antibodies and/or HDV RNA was used to detect hepatitis D. Results: In the years 2005–2022, no confirmed case of acute or chronic HDV infection was reported in Slovakia. The presented survey includes a total of 343 patients, of which 126 were asymptomatic HBsAg carriers, 33 acute hepatitis B, and 184 chronic hepatitis B cases. In a recent prospective study of 206 HBsAg-positive patients who were completely serologically and virologically examined for hepatitis B and D, only 1 anti-HDV IgG-positive and no anti-HDV IgM or HDV RNA-positive cases were detected. In other smaller surveys, two anti-HDV IgG-positive patients were found without the possibility of HDV RNA confirmation. In total, only 3 of 329 HBsAg-positive patients (0.91%) tested positive for anti-HDV IgG antibodies, and none of 220 tested positive for HDV RNA. Conclusion: The available data show that Slovakia is one of the countries with a very low prevalence of HDV infection, reaching less than 1% in HBsAg-positive patients. Routine testing for hepatitis D is lacking in Slovakia, and therefore it is necessary to implement testing of all HBsAg-positive individuals according to international recommendations.

## 1. Introduction

Hepatitis D is the most severe form of chronic viral hepatitis due to the faster progression of the disease to liver cirrhosis and increased risk of hepatocellular carcinoma (HCC). The relative risk of cirrhosis in patients with chronic HBV/HDV infection is 2- to 3-fold compared with HBV mono-infection, the risk of developing HCC is 3- to 6-fold, and the risk of liver decompensation, mortality, and the need for liver transplantation is also 2 times higher [1,2]. After the approval of the first antiviral drug for the treatment of hepatitis D, Bulevirtide, this infection has received increasing attention in recent years [3]. In many countries, testing for the presence of HDV infection in patients with HBV infection is not included among routine standard examinations, and it is implemented rather exceptionally. Two major epidemiological overviews of the prevalence of HDV infection in the world have been published in recent years. Based on a meta-analysis of the prevalence of HDV infection in six WHO regions from 95 countries of the world, the authors estimate the number of infected people worldwide to be 12 million, which corresponds to a global prevalence of 4.5% among HBsAg-positive people. The lowest 3% prevalence was found in the European Region; on the contrary, the African Region and Region of the Americas exhibited almost doubled prevalence [4]. According to the results of another study including data from 83 countries, the pooled prevalence of HDV among HBV carriers is up to 13%, corresponding to 48–60 million infections worldwide. In the monitored European countries, the prevalence ranged from 0.23 to 40.46%, while only three countries reported a prevalence of less than 5% [5]. The true global prevalence of HDV is still unknown due to insufficient data. The estimated prevalence varies substantially between countries and analyses. Some analyses may exclude certain groups at high risk of HDV (i.e., injecting drug users, HIV patients, patients with liver disease). There is also a potential variability in the type of source data between studies. The low proportion of HBsAg-positive individuals being considered for HDV testing is also a major concern [1]. According to the risk factors of parenteral transmission of infection, people at the highest risk are people who inject drugs (PWID), commercial sex workers, men who have sex with men (MSM), HCV- or HIV-infected people, cirrhotics, patients with HCC, and hemodialysis recipients [4]. Geographical differences in prevalence also point to the importance of population migration from endemic countries during the transmission of HDV infection [6]. Relatively few data are available on the incidence of HDV infection in Slovakia; they are completely absent from some regions, and no prevalence data have been published at all so far. Our work aims to summarize all available knowledge, data, and prevalence surveys on the prevalence of HDV infection available in Slovakia, including the results of our most recent prospective study.

## 2. Materials and Methods

### 2.1. Study Design

The work has an exploratory and descriptive character while attempting to capture all eligible patients. We contacted all relevant Infectology and Hepatology centers in Slovakia at which hepatitis D testing was performed in HBsAg-positive patients more or less systematically and not only sporadically in case of clinical suspicion with a request to provide their data. According to our findings, we have verifiable data available about the occurrence of HDV infection from the following sources:Data from the Slovak Public Health Authority on the occurrence of infectious diseases;A pilot project implemented in Eastern Slovakia in 2008 (so far unpublished)—86 patients;Results of the recent B-MARK study (Eastern Slovakia)—206 patients;A set of HBsAg-positive patients examined in Central Slovakia—37 patients;A group of patients with HBeAg-negative chronic hepatitis B followed up in Western Slovakia—14 patients.

According to the available information, it was possible to reliably evaluate the results of 343 patients.

### 2.2. Population Study and Methods

We drew data from the Slovak Public Health Authority from official annual reports on the Analysis of the Epidemiological Situation in the Slovak Republic (SR) for the relevant years available on their website [7].

Patients’ samples examined in the pilot project for HDV prevalence came from a study that was aimed at monitoring the presence of the SEN virus in different groups of the population [8]. A total of 86 residual patient sera were included in this study in the years 2001–2004 with HBV infection and were stored in a frozen state at −80 °C. They were examined in 2008 for anti-HDV IgG antibodies using an ELISA test (Adaltis, Albuccione, Italy).

The prospective observational study, B-MARK, was carried out from 2019 to 2022 in the region of Eastern Slovakia. The aim of this study was a comprehensive study of patients with HBV infection, including an assessment of the importance of individual markers of HBV infection. One of the goals was to determine the prevalence of current or past HDV infection. We obtained selected demographic and epidemiological data from enrolled patients and supplemented laboratory examinations. During the mentioned period, 207 HBsAg-positive patients were included in the study, while 206 of them were also examined for the presence of anti-HDV and HDV RNA. For the qualitative detection of IgG/IgM against hepatitis D virus, a commercial ELISA kit was used (Cusabio, Human hepatitis D virus antibody (IgG/IgM) Houston, TX, USA). For HDV RNA testing, nucleic acid was extracted from patient sera using the QIAamp MinElute Virus Spin Kit (QIAGEN GmbH, Hilden, Germany) according to the manufacturer‘s instructions and stored until use at −20 °C. RNA transcription was performed on a Biometra Tone Thermal Cycler (Analytic Jena, Jena, Germany) at 25 °C for 10 min, followed by 42 °C for 60 min using RevertAid Reverse Transcriptase (Thermo Fisher Scientific, Waltham, MA, USA) and 0.2 µg random hexamers in a reaction mix of 20 µL. cDNA was used in nested PCR with genotyping primers HDV-04/HDV-05 in the 1st reaction and HDV-06/HDV-07 in the 2nd reaction, targeting the ribozyme and HDAg domains of the HDV genome according to Sy BT et al. [9].

The ethics committee of the L. Pasteur University Hospital in Košice approved the B-MARK study on 25 April 2019 (no. 2019/EK/4022). All included patients signed informed consent to participate in the study.

Patients monitored in the region of Central Slovakia had anti-HDV IgG antibodies examined using the qualitative test ETI-AB-DELTAK-2 (DiaSorin S.p.A.) from 2018 to 2023. Patients from Western Slovakia were examined for the presence of HDV RNA in the same time period using a real-time PCR methodology (Bosphore HDV Quantification—Detection Kit v1, Anatolia geneworks, Turkey).

## 3. Results

### 3.1. Slovak Public Health Authority Data

We analyzed data on the number of reported viral hepatitis cases in Slovakia over the period of years 2005–2022. During the monitored 18-year period, 9426 cases of HBV virus infections were reported in Slovakia, of which 6581 were asymptomatic HBV carriers, 1504 were newly diagnosed cases of chronic hepatitis B (CHB), and 1341 cases of acute hepatitis B (AHB) (Table 1). Although we do not have exact data, we know that only a very small proportion of these patients were also tested for the presence of HDV infection. Despite low testing, long-term data show that no confirmed case of acute or chronic HDV infection has been reported during the mentioned period [7].

### 3.2. Pilot Project 2008

In the pilot project in 2008, we examined a total of 86 patients, of which 30 patients were with AHB, 46 with CHB, and 10 were asymptomatic HBV carriers. The majority were men (68.6%), and the average age was 29 years. Seventy out of eighty-six (81.4%) patients also had other epidemiological data related to parenteral transmission of HBV and HDV infections as well as ALT activity at the time of serum collection. The characteristics of this file are shown in Table 2. All patients were examined for the presence of anti-HDV IgG antibodies, while all samples showed a negative result.

### 3.3. B-MARK 2022 Study

The group of 206 examined patients consisted of 90 patients with CHB and 116 asymptomatic HBV carriers. Males predominated in the cohort (61.2%), the average age of the patients was 46.8 years, and the average known duration of infection (since the detection of HBsAg positivity) was 14.6 years. From the perspective of ethnicity, 29 patients (14.1%) were of Roma origin, and 8 (3.9%) came from countries outside of Europe. Of the risk factors for parenteral transmission for HBV and HDV infections, the most frequently identified were tattoos and piercings (15.9%), prison stay (3%), and intravenous drug use (2%). Antiviral treatment with nucleotide(z)ide analogs was used by 62 (68.9%) patients with CHB and none from the group of HBV carriers (Table 3).

Laboratory findings of the monitored patients are shown in Table 4. Most patients (66.5%) had normal ALT activity, and HBeAg positivity was present in 11 (12.2%) patients with CHB. HBV DNA levels depended on the patient group and antiviral treatment. In 186 patients (90.3% enrolled), we also determined liver fibrosis using transient elastography. Liver stiffness values were higher in CHB patients compared to HBV carriers (9.2 vs. 6.0 kPa), as well as the proportion of patients with stage F4 fibrosis (15.3% vs. 1.0%).

Collected sera from all patients were examined for the presence of anti-HDV IgM and IgG antibodies. Anti-HDV IgM positivity was not detected in any sample. Anti-HDV IgG positivity was confirmed in one patient (0.5%). It was a 65-year-old man of Slovak nationality, without known risk factors for parenteral transmission of infection. His HBsAg positivity has been known since 1994 and was HBeAg-negative. In 2009, he underwent treatment with pegylated interferon alfa-2a without a sustained response. Since 2010, he has been using entecavir with a good effect. For a long time, he had normal ALT activity, and HBV DNA values are <13 IU/mL or undetectable. His current liver stiffness value is 10.3 kPa, which corresponds to stage F3 fibrosis according to the Metavir score.

Subsequently, we determined the presence of HDV RNA in all patients using the PCR method. The results were negative in all examined patients, including the only anti-HDV-positive patient described above. Thus, we did not confirm any current HDV infection in the studied set of patients.

### 3.4. HDV Screening in Other Regions

A total of 37 patients, 24 men, and 13 women, were examined in the region of Central Slovakia with an average age of 54.5 ± 13.1 years. According to the diagnosis, there were 24 patients with HBeAg-negative chronic hepatitis B, 10 patients with HBeAg-positive chronic hepatitis B, 1 patient with acute hepatitis B (HBeAg-positive), and 2 patients infected after liver transplantation from an HBsAg-positive and an HBeAg-negative donor. The diagnosis was known for an average of 7.9 ± 5.4 years. 31 patients used antiviral treatment (83.8%). From a laboratory perspective, initial HBV DNA levels varied in the range of 20–700,000,000 IU/mL (median 2000 IU/mL) except for two patients infected after liver transplantation with negative viremia. ALT activity was in the range of 0.25–62.58 µkat/L (average 5.48 µkat/L). The stage of fibrosis was determined in 30 cases, 19 cases with mild to moderate fibrosis F0–F2 (Metavir), 2 with stage F3 fibrosis, and 9 with cirrhosis. Out of 37 examined patients, anti-HDV IgG antibodies were present in 2 cases (5.4%), and 35 patients were negative.

The first anti-HDV IgG-positive patient was a 47-year-old man of Ukrainian nationality, who had HBeAg-negative chronic hepatitis B in the stage of liver cirrhosis treated with tenofovir, slightly increased ALT activity up to 2 times the upper normal limit, and viremia of 482 IU/mL. The patient was lost to follow-up after the initial examination, so HDV infection could not be confirmed by HDV RNA examination.

The second positive patient was a 47-year-old man, an alcohol abuser and intravenous drug user with HBeAg-positive chronic hepatitis B in the stage of fibrosis F0, also with a slightly increased ALT activity up to 2 times the upper normal limit and viremia of 12,800 IU/mL. This patient died because of an esophagus tumor with metastases to the liver, lungs, and vertebrae. HDV RNA testing by PCR could no longer be performed.

Another 14 patients with HBeAg-negative chronic hepatitis B were examined in the region of Western Slovakia. The group consisted of 4 men and 10 women, the average age was 56.4 ± 5.8 years, and the average duration of infection from confirmation of diagnosis was 10.8 years. No patient came from countries outside of Europe or was of Roma origin. We did not confirm known risk factors for parenteral transmission of infection in any of them. Antiviral treatment was used by 13 out of 14 patients. Initial HBV DNA levels ranged from 36–170,000,000 IU/mL (median 3420 IU/mL), and ALT activity was 0.45–11.23 µkat/L (average 2.35 µkat/L). Stage of fibrosis was determined in 13 cases; in 9 cases there was mild to moderate fibrosis F0–F2 (Metavir), in 2 cases stage F3 fibrosis, and in 2 cases cirrhosis. The average liver stiffness determined by transient elastography was 8.5 ± 4.4 kPa. The result of HDV RNA determination was negative in all 14 patients’ samples.

## 4. Discussion

The presented work represents a summary of all currently available relevant data and provides the first-ever information about hepatitis D in Slovakia. Despite the absolute lack of routine testing, the survey can be considered sufficiently representative, and some important conclusions can be drawn from its results.

Out of the total number of 343 enrolled individuals, only 3 out of 329 HBsAg-positive patients tested positive for anti-HDV IgG antibodies (0.91%), and none tested positive of the 220 examined for HDV RNA. We ruled out current HDV infection in one of the three anti-HDV-positive patients using a PCR test; his clinical condition did not indicate this infection either. It was not possible to unequivocally rule out an ongoing HDV infection in the other two patients for objective reasons (death or loss of follow-up), while at least in the Ukrainian patient with chronic hepatitis B in the liver cirrhosis stage, increased ALT, and activity with a viremia of 482 IU/mL, HDV infection could be expected.

Given the established mandatory vaccination against hepatitis B, Slovakia is a country with a low prevalence of HBsAg. The last all-Slovak immunological review from 2018 on 4215 examined individuals (of which 1578 were adults over 19 years of age) showed the overall prevalence of HBsAg in the monitored group to be 0.09% in the group of adults 0.25% [10]. Hepatitis B is not widespread or in the PWID risk group. The only exception is the Roma population, where the HBsAg prevalence is many times higher and reaches up to 12.4% [11]. 

Imported infections can also be an important factor in the possible increase in HDV infection. For example, in a cohort of 1112 HDV-positive patients in France, only 14% were born in France, and the other 21% were in European countries. Almost 2/3 of HDV-positive ones came from countries outside of Europe, and more than half from Sub-Saharan Africa [6]. It is known that the seroprevalence of the hepatitis D virus among HBsAg carriers in Sub-Saharan Africa is particularly high. In West Africa, the overall seroprevalence of the hepatitis D virus in the population of HBsAg-positive persons ranges from 7–33% and in Central Africa even up to 25–64% [12].

The number of migrants from non-EU countries who stay in Slovakia has been very low for a long time compared to other countries. Migrants from the most at-risk countries (Near East and Africa) accounted for only 1.23–2.96% of all migrants. Unregistered migrants without temporary or permanent residence in the Slovak Republic are only entitled to acute health care, and they are not routinely tested for viral hepatitis [13]. We assume that all of these causes along with the low prevalence of HBsAg as well as HIV infections are the reasons for the very low to almost zero incidence of hepatitis D in Slovakia, even in the more at-risk Roma population [7].

Southern and Western European countries as well as other non-European countries report HDV prevalence among HBV carriers usually higher than 5% [5]. Published reports on the prevalence of HDV in Slovakia’s neighboring countries, however, are relatively rare and often come from an earlier period. For example, the prevalence of HDV in HBsAg carriers in Poland based on 17-year-old and older data was 3.52%, while the prevalence of HBsAg in the population is about 1% [5]. Similarly, an approximately 30-year-old study from Hungary of hepatitis D virus markers documented a 13.6% prevalence of hepatitis delta infection in 118 hepatitis B virus seropositive patients suffering from histologically confirmed chronic liver disease, whereas active replication of the virus was proven in six patients [14].

Recent data from a large multicenter study in Austria have been published. The survey was conducted in 10 centers over a period of more than 10 years. The authors reported <1% of confirmed HDV viremic cases among HBV patients, but the real prevalence could be potentially underestimated [15].

The country with the most similar epidemiological situation to Slovakia due to the history of a common state and common socioeconomic and cultural development is probably the Czech Republic. Data from the Czech Republic are also rare, and updated reports are missing. In a review article from 2005, it is stated that hepatitis D occurs in the Czech Republic only rarely. At the time of publication of the paper in the National Reference Laboratory for Viral Hepatitis, National Institute of Health, Prague, HDV infection was proved only in five patients in the last 10 years. Two of these patients were foreigners, one was probably infected during a stay abroad, and only two patients had negative epidemiologic and travel history [16]. Similar to Slovakia, no cases of confirmed hepatitis D were reported in the Czech Republic in 2013–2022, according to the Czech National Institute of Health [17].

One of the lowest prevalences of HDV infection in Europe and worldwide was observed in Slovenia. Of the 1305 HBsAg-positive patients screened, total anti-HDV antibodies were detected in only three patients (0.23%); two of them had an ongoing chronic HDV infection [18]. Slovenia, like Slovakia, is a country with a small population and a low prevalence of HBsAg in the population. With the observed low prevalence of HDV infection, it is estimated that there would be only a few dozen positive people living in both countries.

The latest EASL 2023 guideline recommends anti-HDV testing of all HBsAg-positive individuals with re-testing at any time when superinfection is clinically suspected and testing all anti-HDV-positive individuals for HDV RNA [2]. Some authors have expressed doubt as to whether it is cost-effective to test all HBsAg-positive patients in countries with a very low prevalence of HDV infection [18]. Alternatively, the approach suggested by the slightly older AASLD guideline from 2018 could be used in these countries. This recommends testing only for HBsAg-positive patients with risk factors for HDV infection or patients with low or undetectable HBV DNA and concomitant high ALT activity [19].

On the contrary, the majority of countries currently, even in view of the conducted surveys of the prevalence of HDV infection, are inclined to the opinion that the incidence of hepatitis D is significantly underestimated. One possible solution that would lead to a significant increase in the identification of HDV-positive patients is the introduction of a reflex anti-HDV testing program in all HBsAg-positive patients, similar to the one successfully tested in the Spanish study. The results of this study confirmed not only a significant increase in the absolute number of anti-HDV-positive patients detected but also showed that up to 60% of them had no known risk factors for HDV infection, emphasizing the contribution of reflex texting [20]. 

The main reasons for the lack of screening for HDV infection in Slovakia include the poor availability of anti-HDV antibody testing in routine practice, the lack of standardized tests, the low availability of HDV RNA determination, the lack of effective treatment until recently, as well as the lack of clear international recommendations for hepatitis D screening. Last but not least, the relatively low awareness of this infection among healthcare workers and the public also plays a role. Very similar conclusions were expressed by the authors of recent work from the Czech Republic. As in Slovakia, at present, there is no routine screening for HDV in people with chronic HBV infection in the Czech Republic [21].

The inconsistency of the methodology used in the different subsets studied can be considered one of the limitations of this study. On the other hand, all examinations were performed in certified laboratories of large teaching hospitals, which guarantees the reliability of the results obtained.

## 5. Conclusions

Data obtained from various relevant sources in Slovakia indicate a very low incidence of hepatitis D in the general population. We can consider the low prevalence of chronic HBV infection; an effective program of compulsory vaccination against hepatitis B; a low number of migrants, mainly from countries outside Europe; as well as a favorable epidemiological situation in risk groups of the population, such as IVDU or HIV-positive patients, as key factors in this situation.

Nevertheless, it must be stated that testing for hepatitis D is absolutely insufficient in Slovakia. To improve the detection of hitherto undiagnosed patients, it is necessary to implement routine testing of all HBsAg-positive patients for anti-HDV antibodies, as recommended in international guidelines. The main emphasis in testing must be placed on persons at high risk of parenteral transmission of infection and on migrants from hepatitis D endemic countries.

## Figures and Tables

**Table 1 viruses-15-01695-t001:** Number of reported HBV infections in Slovakia in 2005–2022 [7].

	AHB	CHB	HBV Carriers	All HBV Infections
2005	124	32	346	502
2006	123	28	402	553
2007	103	55	476	634
2008	112	72	374	558
2009	139	99	384	622
2010	112	101	363	576
2011	83	77	382	542
2012	73	85	413	571
2013	74	121	243	438
2014	85	107	449	641
2015	65	134	469	668
2016	50	115	370	535
2017	52	88	424	564
2018	48	88	359	495
2019	49	93	425	567
2020	18	72	214	304
2021	10	66	238	314
2022	21	71	250	342
Together	1341	1504	6581	9426

AHB = acute hepatitis B, CHB = chronic hepatitis B.

**Table 2 viruses-15-01695-t002:** Characteristics of analysis of patients from the pilot project in 2008.

	AHB N = 30	CHB N = 46	HBV Carriers N = 10	Total N = 86
men (%)	18 (60.0)	37 (80.4)	4 (40.0)	59 (68.6)
age (years) ± SD	25.5 ± 10.7	29.6 ± 11.1	36.6 ± 12.4	29.0 ± 11.6
additional data	n = 30	n = 36	n = 4	n = 70
mean ALT (µkat/L)	36.97	3.21	0.86	17.54
normal ALT (%)	1 (3.3)	6 (16.7)	3 (75.0)	10 (14.3)
history of surgery (%)	8 (26.7)	10 (27.8)	1 (25.0)	19 (27.1)
parenteral procedure (%)	4 (13.3)	4 (11.1)	0	8 (11.4)
blood transfusion (%)	1 (3.3)	3 (8.3)	1 (25.0)	5 (7.1)
IVDU	0	0	0	0
tattoo (%)	6 (20.0)	1 (2.8)	0	7 (10.0)
piercing (%)	3 (10.0)	0	0	3 (4.3)
traveling outside the EU (%)	0	3 (8.3)	1 (25.0)	4 (5.7)

AHB = acute hepatitis B, CHB = chronic hepatitis B, IVDU = intravenous drug user, EU = Europe.

**Table 3 viruses-15-01695-t003:** Characteristics of the group of patients from the B-MARK study in 2022.

	CHB N = 90	HBV Carriers N = 116	Total N = 206
men (%)	58 (64.4)	68 (58.6)	126 (61.2)
age (years) ± SD	47.9 ± 11.6	46.0 ± 9.2	46.8 ± 10.3
origin outside the EU (%)	3 (3.3)	5 (4.3)	8 (3.9)
Roma origin (%)	16 (17.8)	13 (11.2)	29 (14.1)
BMI ± SD	27.1 ± 4.3	27.0 ± 4.9	27.0 ± 4.7
duration of HBV infection (years) ± SD	15.1 ± 11.8	14.2 ± 11.6	14.6 ± 11.7
antiviral treatment (%)	62 (68.9)	0	62 (30.1)
additional data	N = 89	N = 112	N = 201
IVDU (%)	4 (4.5)	0	4 (2.0)
tattoo or piercing (%)	18 (20.2)	14 (12.5)	32 (15.9)
imprisonment (%)	4 (4.5)	2 (1.8)	6 (3.0)
on hemodialysis (%)	1 (1.1)	0	1 (0.5)
homosexual orientation (%)	2 (2.2)	1 (0.9)	3 (1.5)

CHB = chronic hepatitis B, IVDU = intravenous drug user, EU = Europe.

**Table 4 viruses-15-01695-t004:** Laboratory findings of patients from the B-MARK study in 2022.

	CHB N = 90 *	HBV Carriers N = 116	Total N = 206
mean ALT (µkat/l) ± SD	1.12 ± 2.07	0.52 ± 0.24	0.78 ± 1.41
normal ALT (%)	49 (54.4)	88 (75.9)	137 (66.5)
HBeAg positivity (%)	11 (12.2)	0	11 (5.3)
median HBV DNA (IU/mL)	58	2779	1282
transient elastography	n = 85	n = 101	n = 186
**liver stiffness (kPa) ± SD**	9.2 ± 9.0	6.0 ± 2.4	7.5 ± 6.5
**stage F4 (%)**	13 (15.3)	1 (1.0)	14 (7.5)

* 62 of CHB patients were taking antiviral therapy; CHB = chronic hepatitis B.

## Data Availability

Data is contained within the article. More detailed data presented in this study are available on request from the corresponding author. The data are part of a database containing broader data not related to this work and personal data of patients, also mainly in the Slovak language.

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
