# Peer review of "Is Slovakia Almost a Hepatitis D Free Country?"

_viruses, 2023, doi:10.3390/v15081695_

Round 1
Reviewer 1 Report
The work is very detailed, and data is critically appraised. I have some additional comments that may be added to the discussion:
1) Are the assays used in Slovakia studies invariably accurate? Although the Adaltis assay (as used by Schréter I, et al. Folia Microbiol 2006;51(3):223-8) has been proven accurate (Brichler S, et al. Clin Microbiol. 2014 May;52(5):1694-7), the B-MARK study used the Cusabio assay, which, only after modifying its cutoff, showed a 81.3% sensitivity and 90.9% specificity (Chow SK, et al. Clin Vaccine Immunol. 2016;23(8):732-734). Do the Authors think that the Cusabio assay may have underestimated the prevalence?
2) The number of migrants from non-EU countries who live in Slovakia is very low: can the Authors support this statement with precise figures? Indeed, the B-Mark study had only 4% of tested persons being born outside of the EU. Is the Slovakian policy sufficiently inclusive when it comes, e.g., to unregistered migrants or migrants without health insurance coverage?
3) The Authors propose to apply the 2018 AASLD guidelines, which recommend testing only for HDV only those who are HBsAg-positive at high risk for HDV infection. However, this seems untenable given the fact that applying a reflex testing up to 60% found positive had no risk factors (Palom A, et al. JHEP Rep. 2022 Jul 21;4(10):100547). WHO and AASLD are rapidly moving to universal reflex testing. Besides, universal reflex testing may allow for a better equity, because if testing is limited to high risk persons, this may be stigmatizing and reduce access to care.
English requires minor editing.
Reviewer 2 Report
Hepatitis D is an important underdiagnosed infection worldwide. It is difficult to detect due to lack of diagnostic kits, this work is relevant for its originality in a country with low endemicity of Hepatitis B.
Sugestion to the authors :
1 To Describe the type of study in the methodology
2- To describe if it was a descriptive study.
3- It is not related if was performed the sample calculation to able to know the number of HVB carriers that should be needed to obtain the power of the study
Question to the authors
1- In all the cases that were reported at the Public Health System data base there were not related HDV carrriers at all?
Best Regards
-
